# Changing Dietary Behavior for Better Biodiversity Preservation: A Preliminary Study

**DOI:** 10.3390/nu13062076

**Published:** 2021-06-17

**Authors:** Wajdi Belgacem, Konstadinos Mattas, George Arampatzis, George Baourakis

**Affiliations:** 1Department of Business Economics and Management, CIHEAM-MAICh, 73200 Chania, Greece; baouraki@maich.gr; 2Department of Agricultural Economics, Aristotle University of Thessaloniki, 54124 Thessaloniki, Greece; mattas@auth.gr; 3School of Production Engineering and Management, Technical University of Crete, 73200 Chania, Greece; garampatzis@pem.tuc.gr

**Keywords:** food consumption, environmental pressures, Mediterranean dietary pattern, Western dietary pattern, European dietary pattern

## Abstract

Broadly consumed dietary patterns, such as the European and Western ones, are exerting pressures on biodiversity both in Europe and globally, and shifting toward a sustainable dietary pattern has thus become a must. This paper constitutes a preliminary communication of the results of a research project on the issue. In this study, the pressures of three dietary patterns (European, Western, and Mediterranean) on biodiversity are addressed in terms of land use, water use, greenhouse gas emissions, and eutrophication impact indicators. The environmental impacts are calculated based on a compositional analysis of each dietary pattern and the environmental footprints of the corresponding food groups. Food balance sheets published by the FAO are used as a basis for the compositional analysis, while the environmental footprints of each of the representative food products are retrieved from related life cycle assessment (LCA) studies. The results show that a shift from the European to the Mediterranean dietary pattern would lead to 10 m^2^/capita/day land savings, 240 L/capita/day water savings, 3 kg CO_2_/capita/day reduction in greenhouse gas emissions, and 20 gPO_4_eq/capita/day reductions in eutrophication potential. Likewise, a shift from the Western to the Mediterranean dietary pattern would lead to 18 m^2^/capita/day land savings, 100 L/capita/day water savings, 4 kg CO_2_/capita/day reduction in greenhouse gas emissions, and 16 gPO_4_eq/capita/day reduction in eutrophication potential. Based on these findings, it is clear that this shift is urgently needed as a step toward environmentally sustainable dietary patterns, such as the Mediterranean one, to preserve biodiversity for future generations.

## 1. Introduction

Food production is considered a driver of environmental pressures on biodiversity. Unless actions are taken to reduce multiple anthropogenic pressures, biodiversity is expected to continue declining at an alarming rate [1]. This is a dangerous development because, firstly, biodiversity should be protected for its intrinsic value, and, secondly, its loss can lead to a breakdown in the functioning of the ecosystem as it threatens the safe provision of the so-called “ecosystem services” [2] that maintain the function of food and freshwater and regulate functions of climate and water purification, in addition to its cultural benefits [3]. Moreover, biodiversity loss and climate change are considered to be intertwined issues [4]. In fact, ecosystems are crucial to mitigate and adapt to climate change impacts [5]. As a countermeasure against biodiversity loss, the Fifth Global Biodiversity Outlook 5 (GBO-5) was released in September 2020 by the UN Convention on Biological Diversity (CBD). It is a final report card on progress toward implementing the Convention on Biological Diversity (CBD) Strategic Plan for Biodiversity, agreed in 2010 with a 2020 deadline. Effectively, the strategy contains five well-designed targets, formulated goals termed Aichi Biodiversity Targets [6], starting with Goal A that addresses the underlying causes of biodiversity loss by mainstreaming biodiversity across government and society, to urging a reduction in direct pressures on biodiversity and promoting sustainable use. Moreover, the strategy aims to improve biodiversity’s status by safeguarding ecosystems, species, and genetic diversity, enhancing the benefits to all from biodiversity and ecosystem services, and finally improving implementation through participatory planning, knowledge management, and capacity building. To this date and according to available evidence, the GBO-5 reports that success in the accomplishment of these goals is very limited [7].

In 2010, the Secretariat of the Convention on Biological Diversity (SCBD) identified five key drivers of biodiversity loss and published them in the third GBO report. The direct driver of biodiversity loss is habitat fragmentation, which reduces the species’ capacity to adapt to climate change. Pollution and climate change lead to a massive loss of biodiversity. Increased levels of nutrients combined with overexploitation and unsustainable use can promote invasive alien species growth at the expense of native species. These drivers act together to create multiple pressures, with one pressure exacerbating the impacts of another on biodiversity and ecosystems [8].

Each of these drivers can be subdivided into several associated environmental pressures. Most authors refer to anthropogenic pressures, yet some distinguish a further category of “natural pressures” [9]. For instance, habitat loss is a consequence of the pressures of land use and water use, climate change is a consequence of the pressures of greenhouse gas emissions (e.g., carbon dioxide), and pollution is a consequence of the pressure of eutrophication. There are many factors that cause these environmental pressures; the most important ones are agriculture and food production [10,11,12,13]. Agriculture now uses approximately 50% of the world’s habitable land, divided into two parts: 77% of the agricultural land used for livestock and dairy and 23% used for crops due to the rapid growth of the population that creates pressure on land use through the intensity of agriculture [14]. Population trends, dietary habits, technology, and crop production have all played significant roles in influencing land use [15]. Food production has a significant impact on land use, where its effect varies depending on food items; for instance, beef requires 30 million square kilometers of land to produce, while pork and poultry require less than two million square kilometers of land for each [16]. Concerning water use, food production is the largest water consumer, with up to 70% of all freshwater allocated for human use and up to 90% in some developing countries [17].

The decline of Arctic sea ice is affecting biodiversity around and beyond the entire biome. The associated ocean acidification pressure, resulting from higher carbon dioxide concentrations representing greenhouse gas emissions (GHG), is also already being observed [18]. One of the reasons for the increase of these gases is food production, which represents the major driver of greenhouse gas emissions [19] as it produces approximately 26% of greenhouse gas emissions in the global GHG emissions, including GHGs from the production of livestock, which represents the most considerable part, with 31% of food production [20]. Most of the recent articles that have studied the impact of food production on GHGs indicate that the production of animal-based foods is associated with higher GHG emissions than plant-based foods [21].

Pollution from nutrients and other sources is a continuing and growing threat to biodiversity in terrestrial, inland water, and coastal ecosystems [8]. One of the most important causes of pollution of water and the ecosystem is eutrophication [22], in other words, when the water becomes enriched with nutrients (nitrogen and phosphorus) [23]. The oversupply of nutrients in fresh and marine water bodies presents a serious ecosystem threat due to impacts on water quality through eutrophication. It is considered that the spread of nitrogen and phosphorus from agricultural production systems is the main driver of eutrophication [24]. Agriculture represents 78% of global eutrophication [20]. The effect of food production varies according to the different food products; for example, red meat generates the highest eutrophication emissions compared to other products [25].

Hence, the food system, including agriculture and food production, has a remarkable impact on biodiversity through food products’ pressures on land use, water use, greenhouse gas emissions (GHG), and eutrophication potential. Furthermore, every food product varies widely in its environmental footprints. Plant-based foods exert the smallest pressure on the environment, unlike animal-based food [26]. Thus, the impact of dietary patterns on the environment can vary depending on the food products consumed.

Recent studies have shown that some dietary patterns, such as the Mediterranean one, are based on four health and nutrition sustainability benefits: low environmental impacts, richness in biodiversity, high sociocultural food values, and positive local economic returns [27,28,29]. UNESCO recognized the Mediterranean diet as an intangible heritage of humanity in 2010 [30]. It is a plant-based diet obtained from a heritage of exchanges over millennia among the people and the different cultures of the Mediterranean region. It consists of more vegetables, fruits and fish, eggs, less meat, frequent intake of cheese, and moderate wine intake with meals. Olive oil and nuts are the main staples of the Mediterranean diet [31]. Unlike the Western dietary pattern, which is generally characterized by high intakes of red meat, processed meat, pre-packaged foods, refined grains, candy and sweets, butter, fried foods, eggs, high-fat dairy products, potatoes, corn, and high-sugar drinks [32], this diet contains about 2200 calories per day, with 50% of calories coming from carbohydrates, 15% protein, and 35% fat [33]. Most studies link the Western diet to heart and health risks. It can easily impair appetite control in humans, an impact that may cause consumers to overeat [34]. Regarding dietary patterns in Europe, particularly in southern regions, people are optimized for a high-plant diet. They actually consume large amounts of fruits and vegetables, which provide plenty of vitamins, minerals, and fiber. Red meats and chicken are used sparingly. The northern European diet is generally high in protein, primarily from meat and dairy products. The diet tends to be low in whole grains, fruits, and vegetables. On the other hand, the diet of Central Europe is strongly influenced by the local climate and seasonal food variations. Pork is the predominant choice of meat in the moderate climates of Central Europe. The extensive use of lard and butter for cooking has made the Central European diet very fatty [35,36].

This paper presents an analysis of the sustainability of the Mediterranean dietary pattern and an assessment of its impact on biodiversity, through the pressure indicators of land and water use, greenhouse gas emissions, and eutrophication potential, by comparing it with the current Western and European dietary patterns.

## 2. Materials and Methods

To address the issues outlined in the introduction and proceed with the sustainability analysis of the three patterns, a construction of scenarios for environmental comparisons of dietary patterns is created. The design of dietary pattern scenarios is incredibly important for testing their environmental impact, and it all starts with the simple idea that varying the quality and amount of food consumption might lead to varied environmental externalities. First, the dietary composition of each dietary pattern is performed, followed by a characterization of the corresponding food groups in relation to the environmental footprints. This study’s approach is largely based on Blas et al. [37]. Figure 1 shows a schematic summary of the study design.

### 2.1. Dietary Composition Scenarios

In the study, three scenarios—the European, Western, and Mediterranean dietary patterns—have been designed to evaluate different dietary habits, all of which ensure the required calorie intake, by mixing different types of foods and calculating the right substitutions among portions.

The current European dietary pattern data were estimated based on the FAO food balance sheet, taking the average food consumption per person per day (g) between years 2008 and 2018. The current Western dietary pattern was represented by the USA food pattern, and data were also estimated based on the FAO food balance sheet during the same period. These data were collected using the FAOSTAT and Statista [38,39]. Food consumption from the Mediterranean dietary pattern data was obtained based on a study conducted by Davis et al. on average food consumption per person per day (g) with the Mediterranean diet [40]. The food composition of the Mediterranean dietary data was adjusted based on Sinkko et al. [41] to make it comparable with the dietary patterns studied. The total energy intake from the different patterns was calculated through food composition tables and stated as comparable at around 2000 kcal (the standard daily dietary intake).

The consumption data were subdivided into 12 food groups to enable a comparison of the dietary patterns, namely, meat, fish, dairy products, eggs, cereal-based products, sugar, oils, tubers, vegetables, legumes, fruits, and nuts. This classification was presented in the literature [40]. Furthermore, the food products with the largest apparent consumption in terms of mass and economic value were chosen as representative products for these groups, based on a study conducted by Crenna et al. [42]—for instance, meat products (including pork, beef, poultry), fish products, dairy products (including milk, cheese, and butter), eggs products, cereal-based products (including bread, pasta, and rice), sugar products, oils (including olive oil and oil products), tubers (including potatoes), vegetables (including tomatoes, onions, and other vegetables), legumes, fruits (including apple, orange, banana, and other fruits) and nut products.

The results from the dietary composition scenarios are presented in Figure 2 that describes the average food consumption of each dietary pattern per day per person per g within the selected food group and their representative products. This highlights the significant disparity in average food intake across the dietary patterns in terms of quality and quantity; for instance, the most common disparity is that the European and Western dietary patterns mainly have a high intake of meat, particularly beef and pork, and a small margin for fruit and vegetables, while the Mediterranean diet is characterized by a high intake of fruit and vegetables and a low intake of meat.

### 2.2. Environmental Footprints

Data regarding the environmental footprints of food products are based on the largest meta-analysis of food system impact studies to date, in which the authors derived the data “from a comprehensive meta-analysis, identifying 1530 studies for potential inclusion, which were supplemented with additional data received from 139 authors. Studies were assessed against 11 criteria designed to standardize methodology, resulting in 570 suitable studies with a median reference year of 2010. The data set covers 38,700 commercially viable farms in 119 countries and 40 products representing 90% of global protein and calorie consumption” [20].

Land use is set as the global average land used to produce one kilogram of different food products, measured in square meters per kilogram. Water use is set as the global average freshwater withdrawals to produce one kilogram of different food products, measured in liters of freshwater per kilogram of the food product. Greenhouse gas emissions are set as the global average greenhouse gas emissions (GHG) to produce one kilogram of different food products, measured in kilograms of CO_2_ equivalents. Eutrophication potential is set as the global average eutrophication emissions to produce one kilogram of different food products, measured in grams of phosphate equivalents. Table 1 summarizes the characterization analysis and provides a comparison of the environmental footprints of food products.

## 3. Results

The results of the analysis are summarized in Table 2. The superiority of the Mediterranean dietary pattern over the other patterns is evident as it has the lowest environmental impact in all pressure indicators. Specifically, this superiority is evident through low land consumption, low contribution to greenhouse gases, and eutrophication potential. The data also show that the Mediterranean dietary pattern is more efficient than the Western and European dietary patterns in terms of water use. A shift toward the Mediterranean dietary pattern instead of current dietary patterns in Europe and the USA can reduce land use to 41% in Europe and 55% in the USA, water use to 18% in Europe and 2% in the USA, greenhouse gas emissions to 36% in Europe and 44% in the USA, and eutrophication potential to 36% in Europe and 31% in the USA.

The contribution of the top food products having the highest footprint within the dietary patterns is presented in Figure 3. By comparing Table 2 to Figure 3, it appears that the impact of dietary patterns on the environment varies depending on the type and quantity of food products consumed. In terms of land use, the Western diet proved to be the most impactful. This may be explained by the fact that the Western diet has higher consumption of beef and cheese (Figure 2), which have the biggest impact on land, followed by the European and lastly the Mediterranean dietary pattern, which proved to be the least impactful in the same context among the three diets studied. It is also illustrated that milk has a high contribution in land use among the three dietary patterns. Furthermore, in the context of water use, the European dietary pattern was found to be the most influential. This could be explained by the fact that the European dietary pattern includes an immoderate amount of food consumed, such as pork, milk, fish (farmed), and cheese, which have a significant impact on water use. For the Western dietary pattern, cheese is the main contributor to the increase in water use; finally, the primary cause for the increase in water use is fish (farmed), followed by the high consumption of vegetables and fruits (Figure 2) in the Mediterranean dietary pattern. Regarding GHG emissions, beef was found to be the greatest contributor to emissions with a large difference compared to other food products, followed by cheese and milk in the Western and European dietary patterns. However, cheese and milk are playing a major role in increasing emissions in the Mediterranean dietary pattern. However, the Mediterranean dietary pattern stays less impactful in the context of GHG emissions compared to the two other patterns studied due to its moderate consumption of the mentioned food products. Finally, in terms of eutrophication, the European dietary pattern is found to be the most impactful diet, with fish and beef being the products with the most influence, followed by the Western diet, which is characterized by heavy consumption of beef. In the Mediterranean dietary pattern, which has the smallest impact when it comes to eutrophication, fish is the product which plays a significant role.

## 4. Discussion

Most of the studies that link food and biodiversity have shown that global dietary patterns should move toward more plant-heavy diets to support biodiversity [43,44]. Moreover, dietary patterns with less consumption in meat could reduce the intensification and expansion of agriculture, therefore reducing the environmental pressures [45].

In this study, it is shown that the Mediterranean dietary pattern, which is based on a plant-based diet, represents the lowest values of the biodiversity pressure indicators, compared with the European and Western dietary patterns (Table 2). Indeed, Implementing the Mediterranean dietary pattern instead of current dietary patterns in Europe and the USA can reduce land use to 41% in Europe and 55% in the USA, water use to 18% in Europe and 2% in the USA, greenhouse gas emissions to 36% in Europe and 44% in the USA, and eutrophication potential to 36% in Europe and 31% in the USA. The main finding aligns with previous studies [46,47,48,49,50] that highlight the benefits of the Mediterranean dietary pattern, which is recognized as a sustainable diet that protects biodiversity. Furthermore, the FAO and Italy have previously highlighted the importance of this dietary pattern for its environmental sustainability [51]. The values concerning the environmental footprints of dietary patterns found in various studies are complex to compare, with our results showing that these studies used different input variables. The use of the environmental footprints of food product data from one source is a major limitation of this research. However, the current study’s findings are aligned with the majority of the available literature that compares the Mediterranean dietary pattern with different dietary patterns [52,53,54,55]. For instance, a study conducted in Spain [56] showed that increasing adherence to the Mediterranean dietary pattern in Spain will reduce land use by 58%, water use by 33%, and greenhouse gas emissions by 72%. Furthermore, a study also conducted in Spain demonstrated that better adherence to the Mediterranean dietary patterns was associated with lower land use 71%, water use 58.88%, and GHG emissions 73% [57]. Another study in Germany tested the environmental footprints of six different dietary patterns, including the current average Dutch diet, the official “recommended” Dutch diet, semi-vegetarian, vegetarian, vegan, and Mediterranean diet [58] and found that the Mediterranean dietary pattern had a high sustainability score compared to the others.

The results demonstrated that the European and Western dietary patterns, with high consumption of animal products, have huge impacts on the environment (Figure 2).This result supports previous findings that indicate that the environmental footprint is highly dependent on the type and amount of food products consumed as the increase in dietary footprints is highly affected by the consumption of animal products [59,60]. Moreover, a study found that plant-based foods have the smallest impact on the environment, unlike animal-based food [61]. This is congruent with the study results that demonstrated that the Mediterranean dietary pattern based on plant-based food has the lowest environmental impact among dietary patterns. In this context, recent studies indicate that food products vary significantly in their environmental footprints. For instance, meat and dairy products were found to be the most relevant food products contributing to the total footprint of dietary patterns [62,63]. Beef production contributes heavily to biodiversity loss, which has disastrous consequences for the environment. Specifically, beef represents the biggest contribution to the dietary patterns’ footprints [64,65,66], as shown in this study, where beef played an efficacious role in the increase in dietary patterns’ footprints. The Western dietary pattern in particular, which is characterized by a high consumption of beef, was found to have the greatest environmental impact. Furthermore, dairy products such as cheese and milk contributed significantly in terms of environmental footprints [67,68,69]. These considerations are in line with the results (Figure 3). Farmed fish was also found to have a notable environmental contribution in all dietary patterns in terms of water use and eutrophication potential [70,71].

The Mediterranean dietary pattern, being a plant-based diet, can protect the environment from further losses and can therefore play an important role in supporting biodiversity [72]. With the alarming pace of food biodiversity loss, shifting toward environmentally friendly dietary patterns can be seen as urgently needed [73]. However, following such a diet does not depend largely on consumer choices, but also on the food system that influences consumers’ preferences. A food system approach helps to show how shifts to sustainable diets can occur without focusing mainly on personal motivations to make food choices [74]. For such shifts in the diet to take place, the food system needs to provide healthy food choices, which are culturally acceptable, affordable, accessible, sustainable, and sufficient for all people [75]. Strategies may change the existence of an unsustainable food system and provide better access to a sustainable diet for the sake of biodiversity. Additionally, system changes may affect consumer choices in more sustainable directions. Therefore, changing habits will also alter aspects of the food systems, and this will create space to change the dietary patterns. The problem is not only the individual level of consumption but primarily the entire food system, which directs consumers in a specific direction, as well as access to healthy foods. Changing what is considered normal by recognizing and modifying the signposts will lead to different consumer choices that favor both personal health and biodiversity.

## 5. Conclusions

This study focuses on the assessments of biodiversity pressures due to the impact of dietary patterns. The results show that shifting toward the Mediterranean dietary pattern exerts less pressure on biodiversity, including lower land use, water use, greenhouse gas emissions, and eutrophication emissions. The amounts of animal foods in dietary patterns, especially meat and dairy products, place significant pressures on biodiversity, with beef products being the most threatening followed by cheese, unlike plant-based food that puts the least pressure on biodiversity, with fruit and vegetable products considered the best, given their low-pressure effects. The current European and Western dietary patterns are exerting pressures on biodiversity both in Europe and globally. However, adopting the Mediterranean dietary pattern instead of the current dietary patterns of Europe and the USA can lead to a significant reduction in the pressure on biodiversity.

We can conclude that it is urgently needed to take steps toward environmentally sustainable dietary patterns such as the Mediterranean dietary pattern. Nonetheless, adopting such a dietary pattern depends not only on consumer decisions but also on the food system that affects consumers’ preferences.

The future directions of this study include extended research on the effect of dietary patterns on biodiversity. Taking into consideration other indicator factors, we look forward to exploring the alternative solutions that can be implemented in different parts of the world to minimize the environmental impact of dietary patterns. This research aims to contribute to understanding the link between dietary patterns and biodiversity, as this specific topic has not been well discussed, as well as to increase awareness of the importance of dietary behavior in biodiversity preservation.

## Figures and Tables

**Figure 1 nutrients-13-02076-f001:**
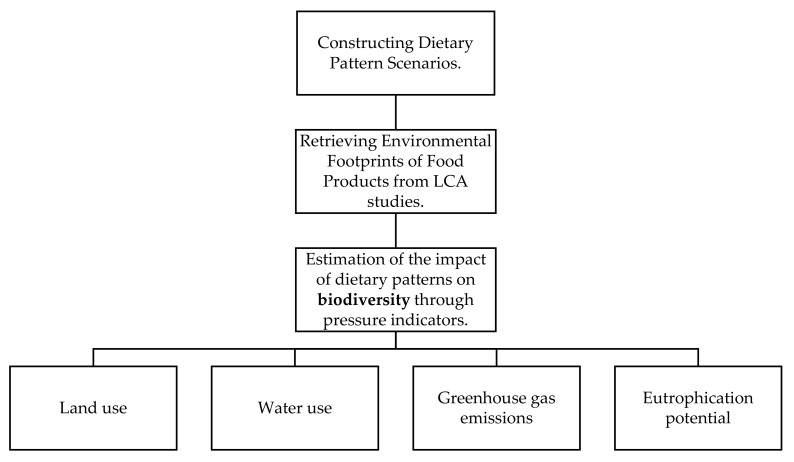
Graphical scheme of design study.

**Figure 2 nutrients-13-02076-f002:**
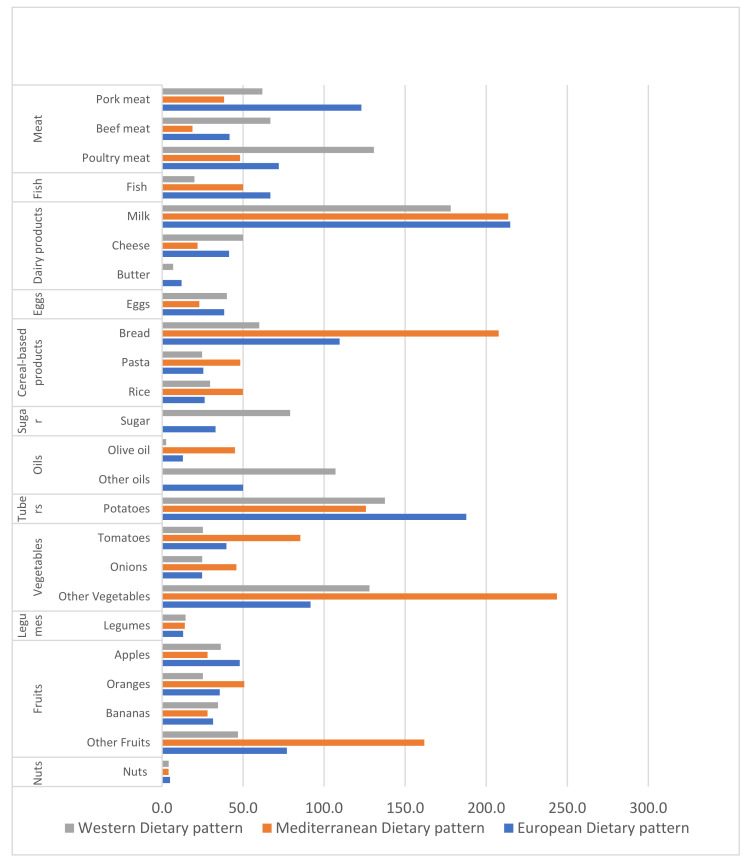
Description of the average food consumption scenarios used in this study per day per person per g, based on 2000 kcal person−1 day−1.

**Figure 3 nutrients-13-02076-f003:**
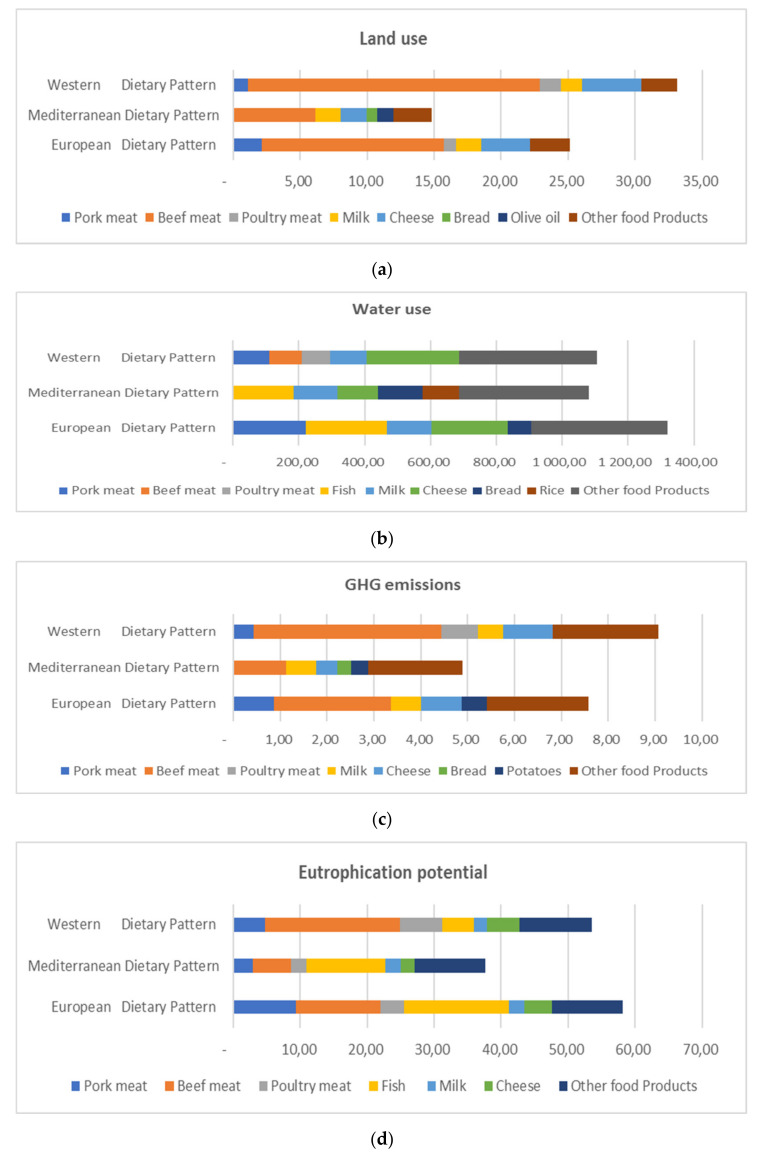
Environmental footprints of (**a**) land use, (**b**) water use, (**c**) GHG emissions, and (**d**) eutrophication potential, of the top food products with the highest footprint within the dietary patterns.

**Table 1 nutrients-13-02076-t001:** Environmental footprints of different food products (global average) in terms of land use, water use, GHG emissions, and eutrophication potential. Colors indicate environmental footprints, from low—green, to light—green, medium—orange, and high—red.

Product Group	Representative Product	Land Use (m^2^/kg)	Water Use (L/kg)	GHG Emissions (kg CO_2_eq/kg)	Eutrophication Potential (gPO_4_eq/kg)
Meat	Pork meat	17.36	1796.00	7.00	76.38
	Beef meat	326.21	1451.00	60.00	301.41
	Poultry meat	12.22	660.00	6.00	48.70
Fish	Fish (farmed)	8.41	3691.00	5.00	235.12
Dairy	Milk	8.95	628.00	3.00	10.65
	Cheese	87.79	5605.00	21.00	98.37
	Butter	2.74	4300.00	11.00	124.50
Eggs	Eggs	6.27	578.00	4.50	21.76
Cereal-based products	Bread	3.85	648.00	1.40	7.16
	Pasta	3.85	648.00	1.40	7.16
	Rice	2.80	2248.00	4.00	35.07
Sugar	Sugar	2.04	620.00	3.00	16.92
Oils	Olive oil	26.31	2142.00	6.00	37.26
	Other oils	10.30	416.75	7.00	23.05
Potatoes	Potatoes	0.88	59.00	2.90	3.48
Vegetables	Tomatoes	0.80	370.00	1.40	7.51
	Onions	0.39	14.00	1.20	3.24
	Other Vegetables	0.38	103.00	1.00	2.27
Legumes	Legumes	8.58	327.33	0.70	10.25
Fruits	Apples	0.63	180.00	0.40	1.45
	Oranges	0.86	83.00	0.30	2.24
	Bananas	1.93	115.00	0.70	3.29
	Other Fruits	0.89	154.00	0.50	2.43
Nuts	Nuts	12.96	4134.00	0.30	19.15

**Table 2 nutrients-13-02076-t002:** Environmental footprints for the Mediterranean dietary pattern, European dietary pattern, and Western dietary pattern.

Pressure Indicators	European Dietary Pattern	Mediterranean Dietary Pattern	Western Dietary Pattern
Land Use (m^2^/capita/day)	25.11	14.80	33.15
Water Use (L/capita/day)	1319.090	1079.965	1105.437
GHG Emissions (kg CO_2_ eq/capita/day)	7.59	4.88	9.08
Eutrophication potential (gPO_4_ eq/capita/day)	55.85	35.50	51.60

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
