# Peer review of "Changing Dietary Behavior for Better Biodiversity Preservation: A Preliminary Study"

_nutrients, 2021, doi:10.3390/nu13062076_

Round 1

Reviewer 1 Report

The aim of this study is very good. The manuscript has been well referenced. Study results may add to the existing knowledge. However, the following comments may further enhance the quality of this paper:

  1. The impact indicators need not be capitalized.
  2. One of the keyword selections duplicate the same as in the paper title. Another selection should be re-chosen.
  3. In the methodology section, the authors mentioned 13 food groups have been classified. But, throughout the entire text, only 12 food groups are being studied. Why?
  4. Figure 1 has not been described in good detail. How different are the three dietary patterns?
  5. It may be helpful if the ordering of the food groups is the same for Figure 1 and Table 1.
  6. Take good care of all the superscripts and subscripts throughout the entire text, figures and tables.
  7. “exerting” is a better word for “placing” in Line 350.

Author Response

The aim of this study is very good. The manuscript has been well referenced. Study results may add to the existing knowledge

Author response: Thank you!

  1. The impact indicators need not be capitalized.

Author response: Thank you for pointing this out. The reviewer is correct, and we have changed this matter. This change can be found in the revised manuscript

  1. One of the keyword selections duplicate the same as in the paper title. Another selection should be re-chosen.

Author response: Agree. We have accordingly modified the keyword “Biodiversity preservation “since it duplicates the same as in the paper title by “Food consumption”

  1. In the methodology section, the authors mentioned 13 food groups have been classified. But, throughout the entire text, only 12 food groups are being studied. Why?

Author response: Thank you for pointing this out. We have corrected the mistake, it was just a typo, 12 food groups have been classified not 13 food groups. This change can be found in the revised manuscript: Methodology – Line 180.

  1. Figure 1 has not been described in good detail. How different are the three dietary patterns?

Author response: Agree, therefore, we have modified the description of figure 2* (figures number has been changed since we have added a new figure following the second reviewer comments). This change can be found in the revised manuscript: Methodology-Line 191-198.

  1. It may be helpful if the ordering of the food groups is the same for Figure 1 and Table 1.

Author response: Agree, we have accordingly modified the food groups on the same order in Figure 2* and Table 1. This can be found in the revised manuscript.

  1. Take good care of all the superscripts and subscripts throughout the entire text, figures and tables.

Author response: Agree, we have carefully revised the manuscript.

  1. “exerting” is a better word for “placing” in Line 350.

Author response: Thank you for pointing this out, we have changed that. This change can be found in Line 405.

Reviewer 2 Report

I suggest to authors to change the type of paper from "Article" to "Communication". Also the authors should add in the Title "Preliminary Study" and pointed out in the text that this is a preliminary study and describe the future directions. A graphical scheme of study design shoud be inserted.

2.1. Composition analysis: this subparagraph should be implemented and also the title reconsidered. Consumption Data are teken into account and food grouping. Are Food Composition Database taken into account?

Lines 148-152 should be better explained; major details on food grouping shoud be given.

Data in Table 2 and Figure 2 should be better described.

Results should be better compared to previous studies in literature

The linguistic revision of whole manuscript should be carried out.

Author Response

  1. I suggest to authors to change the type of paper from "Article" to "Communication". Also, the authors should add in the Title "Preliminary Study" and pointed out in the text that this is a preliminary study and describe the future directions. A graphical scheme of study design should be inserted.

Author response: Thank you for bringing this to our attention. Therefore, we have changed the manuscript following the reviewer comment:

  • We have changed the type of paper from ‘Article’ to ‘Communication’. Line 1
  • Updated the title to ‘Changing dietary behavior for better biodiversity preservation: A preliminary study’ and pointed out in the text “This paper constitutes a preliminary communication of the results of a research project on the issue”. Line 13-14
  • Described the future directions: “The future direction of this study includes extended research on the effect of dietary patterns on biodiversity. Taking into consideration other indicator factors, we look forward to exploring the alternative solutions that can be implemented in different parts of the world to minimize the environmental impact of dietary patterns. This research aims to contribute to understanding the link between dietary patterns and biodiversity, as this specific topic has not been well discussed as well as to increase awareness of the importance of dietary behavior in biodiversity preservation’ Line 412-418
  • Inserted a graphical scheme of study design (Figure 1) on page 4

  1. 1. Composition analysis: this subparagraph should be implemented and also the title reconsidered. Consumption Data are taken into account and food grouping. Are Food Composition Database taken into account?

Author response: Thank you for pointing this out. Therefore, we have updated the mention section. This change can be found in the manuscript Line 163-179.

We have changed the title from “Composition analysis” to “Dietary composition scenarios”. And we had taken food composition database into account, The total energy intake from the different patterns was calculated through food composition tables and stated as comparable ranging around 2000 kcal (the standard daily dietary intake.

‘”2.1 Dietary composition scenarios

In the study, three scenarios: the European, western, and Mediterranean dietary pat-terns, have been designed to evaluate different dietary habits, all of which assure the required calorie intake, by mixing different types of foods and calculating the right substitutions among portions.

The current European dietary pattern data was estimated based on the FAO food balance sheet, taking the average food consumption per person per day (g) between years 2008 and 2018. The current Western dietary pattern was represented by the USA food pat-tern, and data was also estimated based on the FAO food balance sheet during the same period. This data was collected using the FAOSTAT and Statista [40-41]. The food consumption of the Mediterranean dietary pattern data is obtained based on a study conducted by Davis et al. about the average food consumption per person per day (g) of the Mediterranean diet [42]. The food composition of the Mediterranean dietary data is adjusted based on Sinkko et al [43] to make it comparable with the dietary patterns studied. The total energy intake from the different patterns was calculated through food composition tables and stated as comparable ranging around 2000 kcal (the standard daily dietary intake).”

  1. Lines 148-152 should be better explained; major details on food grouping should be given.

Author response: Agree, Therefore, we have updated the manuscript. This change can be found in Line 180-190

“The consumption data was subdivided into 12 food groups to enable a comparison of the dietary patterns, namely meat, fish, dairy products, eggs, cereal-based products, sugar, oils, tubers, vegetables, legumes, fruits, and nuts. This classification was presented in the literature [44]. Furthermore, the food products with the largest apparent consumption in terms of mass and economic value were chosen as representative products for these groups, based on a study conducted by Crenna et al. [45]: for instance, meat products (including pork, beef, poultry), fish products, dairy products (including milk, cheese, and butter), eggs products, cereal-based products (including bread, pasta, and rice), sugar products, oils (including olive oil and oil products), tubers (including potatoes), vegetables (including tomatoes, onions, and other vegetables), legumes, fruits (including apple, orange, banana, and other fruits) and nut products.”

  1. Data in Table 2 and Figure 2 should be better described.

Author response: Agree, we have accordingly updated the manuscript. This change can be found in the revised manuscript: Line 266-291

“The contribution of the top food products having the highest footprint within the dietary patterns is presented in Figure 3. By comparing Table 2 to Figure 3, it appears that the impact of dietary patterns on the environment varies depending on the type and quantity of food products consumed. In terms of land use, the western diet proved to be the most impactful. This may be explained by the fact that the western diet has higher consumption of beef and cheese (Figure 2) which have the biggest impact on the land followed by the European and lastly the Mediterranean dietary pattern, which proved to be the less impactful in the same context among the three diets studied. It is illustrated also that milk has a high contribution to land use among the three dietary patterns. Furthermore, in the context of water use, the European dietary pattern was found to be the most influential. This could be explained by the fact that the European dietary pattern includes an immoderate amount of food consumed, like pork, milk, fish (farmed), and cheese, that have a significant impact on water use. For the Western dietary pattern, cheese is the main contributor to the increase in water use; finally, the primary cause for the increase of water use is fish (farmed), followed by the high consumption of vegetables and fruits (Figure 2), in the Mediterranean dietary pattern. Regarding GHG emissions, beef was found to be the greatest contributor to emissions with a large difference compared to other food products, followed by cheese and milk in the Western and European dietary patterns. However, cheese and milk are playing a major role in increasing emissions in the Mediterranean dietary pattern. But it stays less impactful in the context of GHG emissions compared to the two other patterns studied due to its moderate consumption of the mentioned food products. Finally, in terms of eutrophication, the European dietary pattern is found to be the most impactful diet, with fish and beef being the products with the most influence, followed by the western diet which is characterized with heavy consumption of beef, as for the Medi-terranean dietary pattern, which has the least impact when it comes to eutrophication, with fish being the product playing a significant role in the latter.”

  1. Results should be better compared to previous studies in literature

Author response: Thank you for pointing this out. Therefore, we have added other results of previous studies that align with our results and also reinforce the sustainable character of the Mediterranean dietary patterns. This change can be found in the revised manuscript. Line 349-355

“a study conducted also in Spain demonstrated that better adherence to the Mediterranean dietary patterns were associated with lower land use 71%, water use 58,88%, and GHG emissions 73% [62]. Another study in Germany tested the environmental footprints of six different dietary patterns including the current average Dutch diet, 56n the official ’recommended’ Dutch diet, semi-vegetarian, vegetarian, vegan, and Mediterranean diet [63] and found that the Mediterranean dietary pattern had a high sustainability score com-pared to the others.”

  1. The linguistic revision of whole manuscript should be carried out.

Author response: Agree, we have carefully revised the linguistic of the manuscript.

Round 2

Reviewer 1 Report

An excellent job in revising the original manuscript. The revised manuscript is readied for publication acceptance.

Reviewer 2 Report

The authors have improved the manuscript that it is now suitable for publication. Only one I suggest to better carry out the linguistic revision